# ON INCORPORATING SEMANTIC PRIOR KNOWLEDGE IN DEEP LEARNING THROUGH EMBEDDING-SPACE CONSTRAINTS

## ABSTRACT

The knowledge that humans hold about a problem often extends far beyond a set of training data and output labels. While the success of deep learning mostly relies on supervised training, important properties cannot be inferred efficiently from end-to-end annotations alone, for example causal relations or domain-specific invariances. We present a general technique to supplement supervised training with prior knowledge expressed as relations between training instances. We illustrate the method on the task of visual question answering (VQA) to exploit various auxiliary annotations, including relations of equivalence and of logical entailment between questions. Existing methods to use these annotations, including auxiliary losses and data augmentation, cannot guarantee the strict inclusion of these relations into the model since they require a careful balancing against the end-to-end objective. Our method uses these relations to shape the embedding space of the model, and treats them as strict constraints on its learned representations. In the context of VQA, this approach brings significant improvements in accuracy and robustness, in particular over the common practice of incorporating the constraints as a soft regularizer. We also show that incorporating this type of prior knowledge with our method brings consistent improvements, independently from the amount of supervised data used. It demonstrates the value of an additional training signal that is otherwise difficult to extract from end-to-end annotations alone.

## 1 INTRODUCTION

The capacity to generalize beyond the training data is one of the central challenges to the practical applicability of deep learning, and grows as the task considered grows more and more complex. End-to-end training provides a weak supervisory signal when the task requires a long chain of reasoning between its input and output (Glasmachers, 2017; Marcus, 2018; Zador, 2019). A prime example is found in the task of visual question answering (VQA), where a model must predict the answer to a given text question and related image (see Fig. 1). Typical VQA models trained with supervision (questions/images and ground truth answers) tend to capture superficial statistical correlations, rather than the underlying reasoning steps required for strong generalization (Goyal et al., 2016; Agrawal et al., 2018; Teney & van den Hengel, 2016). Prior knowledge that reflects a deeper understanding of the data than the ground truth answers offers an invaluable – although currently ignored – source of information to train models more effectively.

We propose a method to incorporate, in deep learning models, prior knowledge that is specified as relations between training examples. Incorporating knowledge beyond end-to-end supervision is an opportunity to improve generalization by providing high-level guidance complementary to ground truth labels. The fact that two data elements are equivalent, *e.g.* two questions being rephrasings of each other in a question-answering task, provides more information than merely illustrating that they share the same answer. Such a constraint of equivalence exemplifies a high-level, general concept that is much more powerful than a set of examples sharing a label.

Prior knowledge has previously been incorporated into network architectures in multiple ways, *e.g.* by sharing weights spatially in a CNN (Nowlan & Hinton, 1992). Existing approaches are however often task-specific, and more importantly, usually operate in parameter space (Cohen & Welling,

---

Equivalent questions

---

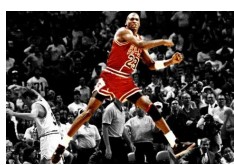 Are the sneakers white, or black? Black.

≡ Are the sneakers black, or white? Black.

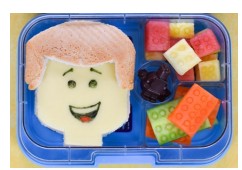 What material is the box made of? Plastic.

≡ What is the rectangular container made of? Plastic.

---

Entailed questions

---

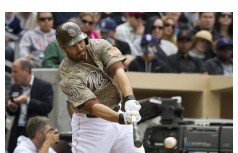 Are there any hats here that are not orange? Yes.

⇒ Is there a hat in the picture? Yes.

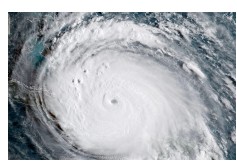 Are there chairs or plates here? No.

⇒ Are there any chairs here? No.

---

Annotations as functional programs

---

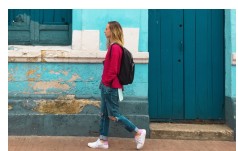 Is the backpack the same color as the house ? No.

```
select:backpack,
select:house,
same:color
```

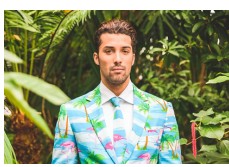 Is the tie brown ? No.

```
select:tie,
verify:color:brown
```

---

Figure 1: We demonstrate our method on the task of visual question answering (VQA), where we exploit three types of auxiliary annotations expressed as relations between training questions. This task-specific knowledge (for example the equivalence of synonymous questions) provides a training signal complementary to the end-to-end annotations of ground truth answers.

2016; Guttenberg et al., 2016; Laptev & Buhmann, 2015; Teney & Hebert, 2016). In contrast, our approach uses knowledge expressed in embedding space, which we find far more intuitive for expressing higher-level knowledge. Indeed, in embedding space, one can specify how the network represents data. In parameter space, one can guide how the network processes these representations. While both can be useful, the former can map more directly to a task- or domain expert's knowledge of the data used. The additional knowledge that we use comes as annotations of relations between specific training instances. We do not require all of the training data to be annotated with this additional knowledge.

Technically, we propose a novel training method that operates in two phases (see Fig. 2). First, we employ the constraints derived from the annotations as soft regularizers. They guide the optimization of the target task to loosely satisfy the constraints. Second, we retrain the earlier layers of the network by distillation (Hinton et al., 2015) using, as targets, embeddings projected on the manifold where the constraints are met perfectly. This second phase is the crux to enforce *hard* constraints on the learned embeddings. A major finding in our experiments is that these hard constraints are more effective the soft regularizers, in all of our test cases.

We present an extensive suite of experiments on synthetic and large-scale datasets (Section 4). In the context of VQA, we apply the method on top of the popular model of Anderson et al. (2018) to leverage three types of annotations illustrated in Fig. 1: relations of equivalence between questions (*i.e.* being rephrasings of one another), of entailment (the answer to a general question being deducible from that of a more specific one), and relations of set membership, where questions are known to share some reasoning steps (*e.g.* questions referring to similar objects or requiring similar reasoning operations). These annotations are provided with the GQA dataset (Hudson & Manning, 2019), but have largely been overlooked due to the difficulty of combining this type of knowledge with end-to-end training. hlWe show that imposing *hard* constraints on linguistic embeddings in this context is superior to the corresponding soft regularizers. We also demonstrate consistent improvements in robustness and accuracy independent from the amount of supervised data, which supports the benefits of such training signals in complement to end-to-end annotations.

The contributions of this paper are summarized as follows.

1. We propose a method to exploit prior knowledge expressed as relations between training instances when training a deep learning model. The method enforces hard constraints on the internal representations learned by the model while allowing end-to-end supervised training, which alleviates issues with soft regularizers.
2. We show that the method is applicable to a range of tasks and types of prior knowledge. We describe its application to three generic types of relations (symmetric/equivalence, asymmetric, and set membership) and show that it does not require domain-specific expert knowledge. In many cases, the specification of constraints in embedding space is more intuitive than the alternative practice of designing regularizers in parameter space.
3. We demonstrate the benefits of the method on the task of VQA. We show how to exploit three types of auxiliary annotations about the training questions (equivalence, entailment, and common reasoning steps). This is the first published VQA model to make use of these annotations, which our method allows us to leverage to bring clear improvements in robustness and accuracy. Our results suggest that they provide a training signal complementary to end-to-end annotations.

## 2 RELATED WORK

**Rules and prior knowledge in neural networks**   Approaches for including rules in machine learning either rely on priors following from Bayes rule, or on regularizers that induce constraints in the parameter space of the model. Hu et al. (2016) used posterior regularization to embed first-order rules. Their objective is to improve the predictions of a learned model to better comply with hand-designed rules describing the task of interest. In comparison, we are interested in embedding instance-level auxiliary annotations, which can be seen as rules applied only to some of the training examples. The objectives of the two methods are distinct and complementary. The major innovation of our method is to enforce *hard* constraints on the learned embeddings, whereas general rules in Hu et al. (2016) are softly balanced with the learned predictions.

Works on differentiable theorem proving (Rocktäschel & Riedel, 2017) and neural reasoning (Peng et al., 2015; Marra et al., 2019) learn vector representations of symbols under constraints such as mutual proximity of similar symbols. General techniques for imposing hierarchical structure on an embedding space include order embeddings (Vendrov et al., 2016) and non-Euclidean representations (Ganea et al., 2018; Nickel & Kiela, 2017). In this paper, we use similar ideas to constrain the embeddings learned for a complex multimodal task (VQA), with the objective of enforcing specific (grounded) constraints, rather than imposing a prior on the overall structure of the embedding space. We propose a novel training procedure to enforce these hard constraints, whereas the above works use soft objectives and regularizers. Many existing works have also described models that enforce known invariances and equivariances (*e.g.* Cohen & Welling (2016); Guttenberg et al. (2016); Laptev & Buhmann (2015); Teney & Hebert (2016)) although they each use an ad-hoc, task-specific design rather than a general technique.

In one of our use cases for VQA, we turn annotations of set membership (program annotations) into linear constraints on the learned embeddings. This bears similarities with (Pathak et al., 2015), where the authors turned image-level tags into linear constraints to train a model for semantic image segmentation. They derive, from the constrained optimization problem, an objective amenable to SGD that is robust to hard-to-enforce and competing constraints. Our contribution, on the opposite, shows how to enforce constraints strictly, which, in our applications, proved superior to a soft-regularized objective.

**Visual question answering (VQA)**   The task of VQA requires a model to answer a previously unseen question expressed in natural language about a previously unseen image (Antol et al., 2015; Goyal et al., 2016; Teney et al., 2017; Wu et al., 2017). The mainstream approach poses VQA as a classification task over a large set of typical short answers, and it uses supervised training with a large training set of questions/answers. We use VQA as a testbed in this paper, because the task exposes flaws of a purely end-to-end, supervised approach (Agrawal et al., 2018; Goyal et al., 2016; Teney & van den Hengel, 2016). The weak signal provided by the answer to a training question makes it difficult for models to capture reasoning procedures that generalize across examples and domains (Chao et al., 2018; Teney & van den Hengel, 2017; 2019). The method proposed in this

paper allows using additional annotations during training. A line of works have used the CLEVR synthetic dataset (Johnson et al., 2016) and its annotations of questions as programs (sequence of reasoning steps to be carried out to find the answer) as an additional source of supervision. These annotations are typically used to learn modular network architectures (Andreas et al., 2016a;b), but these annotations were only possible due to the closed-world, synthetic nature of the dataset. The method we propose is much more flexible than these approaches. It can exploit partial annotations on a subset of the data (*i.e. some* questions have *some* operations in common). It does not associate a particular meaning to operations in the programs, allowing program-like annotations that are not actually executable.

Shah et al. (2019) proposed a generative model of questions conditioned on the answer for VQA. They used it for data augmentation while ensuring that all generated versions lead to the same answer, *i.e.* enforcing cycle consistency. One of our use cases also involves multiple forms of a same question, but these are given in our dataset, whereas Shah et al. (2019) focuses on learning them from the data.

Mao et al. (2019) combined symbolic AI with neural networks for VQA. The system learns a vocabulary of visual concepts, but its operation depends on a pre-specified domain-specific language and manually-designed modules. These contributions are orthogonal to ours. For example, our method could apply to the representations of questions learned in their semantic parser.

**Distillation of neural networks** The original purpose of distillation is to transfer knowledge from a "teacher" model to a "student" that uses smaller computational resources (Bucilua et al., 2006; Hinton et al., 2015; Furlanello et al., 2018). In this paper we use distillation to retrain some layers of a network to produce desired embeddings. The outputs of the teacher are first projected to fit desired constraints, and the projections are then used as targets by the student. The student thereby learns the teacher's knowledge in addition to the prior knowledge represented by the constraints. The method of Hu et al. (2016) mentioned above uses a teacher/student distillation procedure during training. Their best results are however obtained with the teacher network at test time, rather than the student. In our case, the distillation phase is critical to enforce hard constraints.

## 3 PROPOSED APPROACH

### 3.1 OVERVIEW

The intuition behind our approach is that the knowledge we wish to incorporate can serve to shape the space of the learned representations. First, we translate this task-specific knowledge into constraints to be placed on representations learned by the model. This translation does not necessarily require domain knowledge. Our experiments on VQA demonstrate three generic types of constraints (vector equality, norm inequality, and linear programs). As one specific example, the knowledge of equivalence between pairs of questions as in Fig. 1 translates to a constraint of equality of their corresponding embeddings $x_1$ and $x_2$. We then train a network that strictly respects these constraints thanks to a two-phase procedure (see Fig. 2). In the first phase, we optimize the network end-to-end for the target task objective, with a regularizer that encourages a soft version of the constraints (the distance $||x_2 - x_1||^2$ in the above example). We also insert a special operation in the network that projects $x_i$ onto $x_i'$, that lies on the manifold where the constraints are strictly met. The subsequent layers only receive this version $x_i'$ of the embeddings, such that they are optimized to deal with embeddings that satisfy the constraints. The first phase ends once overfitting is detected (early stopping). At this point, the soft regularizer only helped to loosely fit the constraints since the optimization is interrupted before the loss reaches its minimum. The second phase then serves to improve the fit to the constraints, without further overfitting the task objective. In the second phase, the output layers are thus frozen, and we retrain the earlier layers with a distillation objective, using $x_i'$ as targets (which do respect the constraints). Since the output layers are frozen, this second phase can proceed to convergence, *i.e.* until these earlier layers produce embeddings that perfectly fit of the desired constraints.

The following describes each phase formally. Additional details are provided in the appendix.

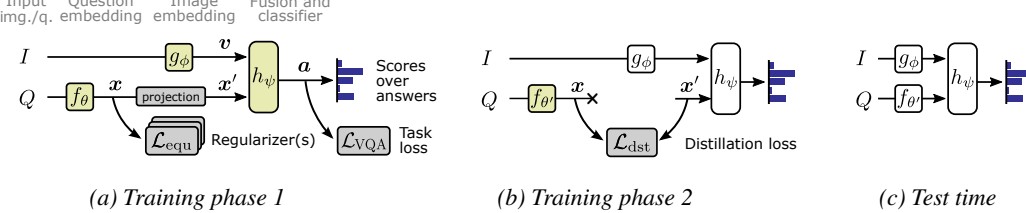

*(a) Training phase 1*      *(b) Training phase 2*      *(c) Test time*

Figure 2: We propose a two-phase training procedure to combine task supervision with hard constraints on learned embeddings. In the first phase, the model is trained with the end-to-end supervision (task loss), regularizers that represent soft versions of the constraints, and an internal projection of the embeddings (see text for details). In the second phase, the embedding layers are retrained with a distillation objective, using the projected embeddings as targets. The trained layers are highlighted.

## 3.2 PHASE 1: TASK LOSS AND SOFT REGULARIZERS

We first train all layers of the network end-to-end. In the model depicted in Fig. 2, these layers make up the functions $f_\theta(\cdot)$, $g_\phi(\cdot)$, and $h(\cdot)$. We use the supervised task loss together with soft regularizers. The task loss for VQA is the binary cross-entropy $\mathcal{L}_{\text{VQA}}(\boldsymbol{a}, \hat{\boldsymbol{a}})$ where $\hat{a}$ is the one-hot encoding of the ground truth answer. The soft regularizers encourage embeddings to respect the constraints defined in the previous section. For the equivalent questions, we use the squared L2 distance: $\mathcal{L}_{\text{equ}}(\boldsymbol{x}_1, \boldsymbol{x}_2) = ||\boldsymbol{x}_2 - \boldsymbol{x}_1||_2^2$. For the entailed questions, we use the difference of $p$-norms: $\mathcal{L}_{\text{ent}}(\boldsymbol{x}_1, \boldsymbol{x}_2) = \max\big(0, ||\boldsymbol{x}_1||_p - ||\boldsymbol{x}_2||_p\big)$. For the annotations of operations, we use the binary cross-entropy on top of a logistic regression that uses the boundary of the half-space associated with $\boldsymbol{o}_k$ as its decision boundary: $\mathcal{L}_{\text{ops}}(\boldsymbol{x}_i) = -\Sigma_k \mathbb{H}\big(\sigma(\boldsymbol{w}_k \boldsymbol{x}_i + b_k); \mathbb{1}(o_k \in Q_i)\big)$. See Table 1 for a summary.

The model is optimized with an objective that combines the task loss with the three regularizers:

$$
\min_{\theta, \phi, \psi, \{\boldsymbol{w}_k\}, \{b_k\}} \Sigma_i \, \mathcal{L}_{\text{VQA}}(\boldsymbol{a}_i, \hat{\boldsymbol{a}}_i)
$$
$$
+ \lambda_{\text{equ}} \, \mathcal{L}_{\text{equ}}(\boldsymbol{x}_i, \boldsymbol{x}_{\text{equ}(i)})
$$
$$
+ \lambda_{\text{ent}} \, \mathcal{L}_{\text{ent}}(\boldsymbol{x}_i, \boldsymbol{x}_{\text{ent}(i)}) \qquad (1)
$$
$$
+ \lambda_{\text{ops}} \, \mathcal{L}_{\text{ops}}(\boldsymbol{x}_i)
$$

where the functions $\text{equ}(i)$ and $\text{ent}(i)$ sample training instances respectively equivalent to, and entailed by the $i$th one. The factors $\lambda_{\text{equ}}$, $\lambda_{\text{ent}}$, and $\lambda_{\text{ops}}$ are non-negative scalar hyperparameters (see Section D). This objective encourages the embeddings to follow the desired constraints in a *soft* manner, balanced with the end-to-end task objective. However, some constraints are known to be exact. Taking the case of equivalent questions as an example, the network must learn identical representations for synonymous questions. We therefore project the learned embeddings $\{\boldsymbol{x}_i\}$ onto $\{\boldsymbol{x}'_i\}$ in such a way that they collectively respect the desired constraints. The projected embeddings are then used by the remaining of the network, *i.e.* $\boldsymbol{a} = h_\psi(\boldsymbol{x}', \boldsymbol{y})$. In this way, the remaining layers are optimized to use embeddings that perfectly respect the constraints, which supports the second training phase described below.

The projection functions are given in Table 1, bottom row. For equivalent questions, we replace their embeddings by their arithmetic mean to make them identical. For entailed questions, we normalize them to the mean of their norms to make the inequality constraint just satisfied. For the program operations, we perform a descent on the gradient of $\mathcal{L}_{\text{ops}}(\boldsymbol{x}_i)$ for a fixed number of steps (see appendix for details). This identifies a local projection $\boldsymbol{x}'_i$ that best respects the constraints resulting from the constituent operations of the question, even when the conjunction of these constraints is not strictly feasible.

## 3.3 PHASE 2: ENFORCING HARD CONSTRAINTS BY DISTILLATION

The second step of our training procedure aims at improving the layers ($f_\theta$) that produce the embeddings such that the constraints are more closely respected. In principle, this could be achieved with the soft regularizers, but in practice, the task loss often converges at a different rate than the regularizers – regardless of the weights assigned to each term in the objective. Indeed, early stopping is usually employed to prevent overfitting before the convergence of the regularizers. A naive, fur-

Table 1: Summary of the constraints, soft regularizers, and projection functions for the three types of annotations we use for VQA. The figures are conceptual representations in a 2D embedding space.

| Equivalent questions $\boldsymbol{Q}_1 \equiv \boldsymbol{Q}_2$ | Entailed questions $\boldsymbol{Q}_1 \Rightarrow \boldsymbol{Q}_2$ | Functional programs (operations) $\boldsymbol{Q}_i \equiv \{\boldsymbol{o}_1, \boldsymbol{o}_2, ..., \boldsymbol{o}_{K_i}\}$ |
|---|---|---|
| Constraints on question embeddings | | |
| $\boldsymbol{x}_1 = \boldsymbol{x}_2$ | $\|\boldsymbol{x}_1\|_p \geq \|\boldsymbol{x}_2\|_p$ 
 (using $p = 1$) | $\wedge_{k:\boldsymbol{o}_k \in \boldsymbol{Q}_i} (\boldsymbol{w}_k \boldsymbol{x}_i + b_k \geq 0)$ 
 $\wedge_{k:\boldsymbol{o}_k \in \mathcal{O} \setminus \boldsymbol{Q}_i} (\boldsymbol{w}_k \boldsymbol{x}_i + b_k < 0)$ |
| Soft regularizers | | |
| $\mathcal{L}_{\text{equ}}(\boldsymbol{x}_1, \boldsymbol{x}_2) =$ 
 $\|\boldsymbol{x}_2 - \boldsymbol{x}_1\|_2^2$ | $\mathcal{L}_{\text{ent}}(\boldsymbol{x}_1, \boldsymbol{x}_2) =$ 
 $\max(0, \|\boldsymbol{x}_2\|_p - \|\boldsymbol{x}_1\|_p)$ | $\mathcal{L}_{\text{ops}}(\boldsymbol{x}_i) =$ 
 $-\Sigma_{k:\boldsymbol{o}_k \in \boldsymbol{Q}_i} \log(\sigma(\boldsymbol{w}_k \boldsymbol{x}_i + b_k))$ 
 $-\Sigma_{k:\boldsymbol{o}_k \in \mathcal{O} \setminus \boldsymbol{Q}_i} \log(1 - \sigma(\boldsymbol{w}_k \boldsymbol{x}_i + b_k))$ |
| Projections to enforce hard constraints | | |
| $\boldsymbol{x}_1' = \frac{(\boldsymbol{x}_1 + \boldsymbol{x}_2)}{2}$ 
 $\boldsymbol{x}_2' = \frac{(\boldsymbol{x}_1 + \boldsymbol{x}_2)}{2}$ | $\boldsymbol{x}_1' = \frac{\boldsymbol{x}_1}{\|\boldsymbol{x}_1\|_p} \cdot \frac{\|\boldsymbol{x}_1\|_p + \|\boldsymbol{x}_2\|_p}{2}$ 
 $\boldsymbol{x}_2' = \frac{\boldsymbol{x}_2}{\|\boldsymbol{x}_2\|_p} \cdot \frac{\|\boldsymbol{x}_1\|_p + \|\boldsymbol{x}_2\|_p}{2}$ | $\boldsymbol{x}_i' = \boldsymbol{x}_i$ then, for $T$ steps: 
 $\boldsymbol{x}_i' \leftarrow \boldsymbol{x}_i' - \alpha \nabla_x \mathcal{L}_{\text{ops}}(\boldsymbol{x}_i')$ |

ther training using the regularizers alone will cause the learned embeddings to diverge from the task objective and performance will decrease. Our solution is to retrain the layers $\boldsymbol{x} = f_\theta(Q)$ by distillation, using the projected embeddings $\boldsymbol{x}'$ as targets (Fig. 2). Practically, the other parts of the network are frozen ($g_\phi(\cdot)$ and $h_\psi(\cdot)$) and we optimize the following objective: $\min_{\theta'} \Sigma_i \mathcal{L}_{\text{dst}} \|\boldsymbol{x}' - \boldsymbol{x}\|^2$, with $\boldsymbol{x} = f_{\theta'}(Q)$. This objective uses an L2 loss with $\boldsymbol{x}'$ as the targets. The key here is to hold these targets fixed during the distillation (*i.e.*, they remain the projection of the embeddings obtained at the end of first phase of training). There is now no risk of overfitting on the target task (not any further than at the time the distillation is started) and the only factor driving an evolution of the network is the objective of closely following the constraints resulting from the projection.

### 3.4 APPLICATION TO VISUAL QUESTION ANSWERING

**Baseline VQA model** The proposed method is generally applicable to a variety of tasks and architectures. We now describe its application to the context of VQA used in our main experiments. We abstract a VQA model as depicted in Fig. 2. The input question $Q$ and image $I$ are passed through the embedding functions $f_\theta(\cdot)$ and $g_\phi(\cdot)$, respectively. The first is typically a word embedding followed by an LSTM or GRU, while the second is typically a CNN or R-CNN feature extractor followed by a non-linear transformation. These embedding layers produce the two vector representations $\boldsymbol{x} = f_\theta(Q)$ and $\boldsymbol{y} = g_\phi(I)$, with $\boldsymbol{x} \in \mathbb{R}^M, \boldsymbol{y} \in \mathbb{R}^N$. These vectors are passed to a fusion and output stage that produces the vector $\boldsymbol{a} = h_\psi(\boldsymbol{x}, \boldsymbol{y})$ containing scores over a large set of candidate answers. Attention mechanisms are contained within $h(\cdot)$ with the given notations, leaving $\boldsymbol{x}$ and $\boldsymbol{y}$ purely unimodal. This general notation encompasses all "joint embedding" approaches typically used for VQA (Wu et al., 2017) and the contributions of this paper are agnostic to the underlying implementation.

**Embedding-space constraints for VQA** We consider three exemplar forms of prior knowledge for VQA that can be used independently or in conjunction. We first describe how to translate this task-specific knowledge to constraints that we will enforce on the question embeddings $\boldsymbol{x}_i$ learned within the model. Refer to Table 1 for a summary.

1. First, we consider relations of equivalence between questions, denoted by $\boldsymbol{Q}_1 \equiv \boldsymbol{Q}_2$. Such questions are rephrasings of one another, using synonyms or linguistic patterns that do not affect their overall meaning (see example in Fig. 1). We naturally translate this to a constraint of equality of the embeddings, that is $\boldsymbol{x}_1 = \boldsymbol{x}_2$.

2. Second, we consider relations of entailment between questions, denoted by $\boldsymbol{Q}_1 \Rightarrow \boldsymbol{Q}_2$. This generally corresponds to a specific question being informative to answer a more general one, *e.g. Is there a blue car in the picture ? Yes. $\Rightarrow$ Is this a car ? Yes.* The major distinction with equivalence relations is that entailment is generally not symmetric. Motivated by works on order embeddings (Vendrov et al., 2016) and hyperbolic networks (Ganea et al., 2018; Nickel & Kiela, 2017), we impose an order on the $p$-norm of the learned embeddings, such that $||\boldsymbol{x}_1||_p \geq ||\boldsymbol{x}_2||_p$. The L1 norm proved the most effective in practice.

3. Third, we consider annotations of functional programs, *i.e.* the reasoning operations involved in each question. A question can be translated to a set of operations $\boldsymbol{Q}_i \equiv \{\boldsymbol{o}_1, \boldsymbol{o}_2, ..., \boldsymbol{o}_{K_i}\}$ with $\boldsymbol{o}_k \in \mathcal{O}$, a large vocabulary of possible operations (see Section D). In order to deal with the variable size of such a definition (the number of operations often depends on the length and complexity of the question), we translate such a definition of a question to the membership of its embedding to the intersection of subspaces associated with its operations (see Table 1 lower-right). The subspaces are learned: for each possible operation $\boldsymbol{o}_k \in \mathcal{O}$, we define its associated subspace as $\{\boldsymbol{x} : \boldsymbol{w}_k \boldsymbol{x} + b_k \geq 0\}$ with a learned vector $\boldsymbol{w}_k \in \mathbb{R}^a$ and scalar $b_k$. Since a question usually involves multiple operations, its embedding is subject to a conjunction of constraints as defined in Table 1.

A practical benefit of our overall approach is its ability to use partial annotations. In other words, not all of the above true relations need to be specified. This is particularly relevant for the functional programs, where only some of the operations of some questions may be annotated.

## 4 EXPERIMENTS

### 4.1 EXPERIMENTS ON A TOY TASK: SEQUENTIAL ARITHMETIC

We first evaluate the method in controlled conditions. We designed a simple arithmetic task in which the model receives a integer digit $x \in [-9, +9]$ as input, together with a variable-length sequence of operations $\{o_1, o_2, \ldots\}$, each one being an addition or multiplication with an integer digit. The desired output is the result of the application of these operations in sequence (*i.e.* without taking into account precedence, see Fig. 3). We build a simple baseline model to learn this task from a supervised dataset (see details in appendix B). This model first maps every token input (digit and operations) to a vector embedding. The operation vectors are passed through a GRU. The final state of the GRU (corresponding to $\boldsymbol{x}$ in Section 3) is concatenated with the embedding of the input digit ($\boldsymbol{v}$) and passed through an MLP with a linear output layer. The model is optimized by SGD for a regression loss (squared L2). Note that the input digit and operations are embedded separately so that we can apply constraints on the embedding of the operations alone.

The proposed method is used on this task as follows. We generated, with our training set, additional annotations of sequences of operations that are known to be equivalent. For example, the training instances $\{3, +1, *2, = 8\}$ and $\{4, *2, -2, +4 = 10\}$ are marked as such, since $(x + 1) * 2 = ((x * 2) - 2) + 4$ for all $x$. These experiments test whether annotations of equivalent sequences are complementary to training examples, where the operations are demonstrated on specific $x$'s. These annotations are naturally translated into constraints of equivalence of the embeddings of operations $\boldsymbol{x}$ as described in Table 1, column 1. We compare in Fig. 3 the effect of our method with a baseline that uses a standard soft regularizer (squared L2 distance) to minimize the distance between the embeddings of equivalent sets of operations (the weight of this regularizer is tuned by cross-validation). Our method proves more effective with all tested amounts of training data and amounts of additional annotations. The data augmentation baseline consists in replacing the set of operations of a training example by another equivalent one, sampled at random from other training examples. This is a particularly strong baseline as seen in Fig. 3, but importantly, it is only relevant for relations of equivalence, while the proposed method has a much wider applicability. Finally, the use of these additional annotations shows significant benefits over a model trained solely from end-to-end supervision across a range of number of training examples. This further supports the overall benefit of leveraging complementary training signals in additional to supervised, end-to-end training.

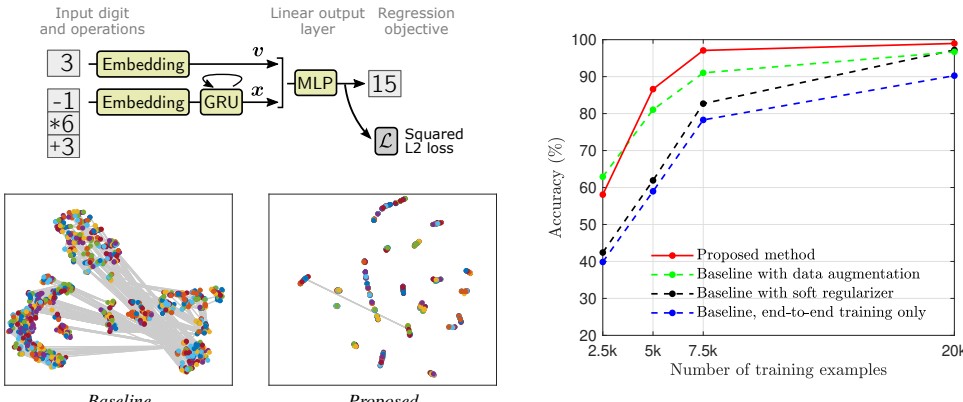

Figure 3: Experiment on a toy task for sequential arithmetic (see Section 4.1). The baseline model (top left) takes in a digit and a sequence of operations, and is trained with supervision to predict the result. We use the proposed method to exploit auxiliary annotations of equivalence between sequences of operations from multiple training examples. This brings significant improvements in accuracy (right plot) over the baseline and over classical techniques that use the same additional annotations. We visualize T-SNE projections (bottom right) of the learned representations ($x$) of sequences of operations from the test set with the baseline (left) and the proposed model (right; 1000 points in both cases). We draw gray lines between the representations of equivalent sets operations: with the proposed method, equivalent representations are virtually identical (only one gray line is visible).

## 4.2 EXPERIMENTS ON VISUAL QUESTION ANSWERING

We conducted an extensive set of experiments to evaluate the proposed method on top of a popular VQA model (the Bottom-Up-Top-Down Attention, BUTD of Anderson et al. (2018); Teney et al. (2018)). We use the GQA dataset (Hudson & Manning, 2019), which contains all three types of annotations discussed in Section 3.4. Details on the implementation and on the data are provided in the supplementary material.

**Ablative evaluation** We first evaluate the three types of annotations in isolation (see Table 2). Optimal loss weights were determined empirically as $\lambda_1$=0.5, $\lambda_2$=0.1, and $\lambda_3$=0.1 for best performance on the validation set. In the following experiments, we set the weights of the losses not used to 0.

- Equivalent questions. We first evaluate a simple baseline to use equivalent questions by data augmentation. We simply replace questions at random during training by any of their equivalent forms (including themselves). The resulting model (row 2) is actually slightly worse than the original one. Training with the proposed soft regularizer, in contrast, provides a significant improvement in accuracy. Adding the projection to further constrain the embeddings has minimal effect on its own, but it allows to retrain the embedding layers by distillation, which provides another clear boost in performance (see also Fig. 3). We compare two variants: a distillation "student" initialized from scratch, or initialized with the weights of the teacher. The former option follows more closely the intent of the original works on distillation (Bucilua et al., 2006; Hinton et al., 2015; Furlanello et al., 2018). Intuitively, a re-optimization from scratch might better exploit the gradients of the distillation loss and lead the SGD through a less circuitous path than the loss used when training the teacher. In practice, both options performed well with a slight advantage for a fine-tuning.

- Entailed questions. We compare two types of soft regularizers. The first, as a baseline, is the same L2 distance as used for the equivalent questions (row 7). The second, our proposed order on L1 norms proves clearly superior (row 8). Entailment relations are asymmetric and this cannot be captured with the L2 distance. We experimented with the reverse order in the proposed formulation (swapping the premise and consequence) and results were similar (not in the table). This further confirms that the key is to model the asymmetry in the relations. We also experimented with alternative options, including order embeddings (Vendrov et al., 2016) and order on L2 norms, neither of which matched the performance of the order on L1 norms. Adding the projection and the distillation has again a positive effect.

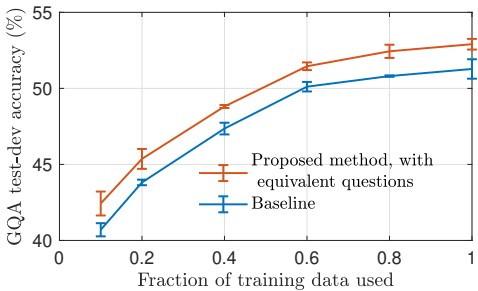

Figure 4: We compare the proposed method with the baseline trained with reduced amounts of data. The absolute improvement over the baseline is consistent and maintained regardless of the amount of training data used. This suggests that the additional information brought in is truly complementary to end-to-end supervision. Model used: row (6) in Table 2; we plot the mean accuracy of four single models trained with different random seeds; error bars represent +/- one standard deviation.

Table 2: Ablative evaluation on GQA (answer accuracy in percents). Every experiment was repeated with four different random seeds. We report the mean and standard deviation of single models, the performance of the ensemble of these four models as in Teney et al. (2018).

| Method | GQA validation | | GQA test-dev | |
|---|---|---|---|---|
| CNN+LSTM (Hudson & Manning, 2019) | 49.2 | – | – | – |
| BUTD, implementation of (Hudson & Manning, 2019) | 52.2 | – | – | – |
| MAC (Hudson & Manning, 2018) | 57.5 | – | – | – |
| (1)  Our BUTD baseline | $58.9 \pm 0.2$ | 61.5 | $51.3 \pm 0.6$ | 54.2 |
| Equivalent questions | | | | |
| (2)  Data augmentation | $58.4 \pm 0.2$ | 61.5 | $51.0 \pm 0.4$ | 53.8 |
| (3)  Soft regularizer | $59.1 \pm 1.1$ | 61.5 | $52.4 \pm 0.7$ | 54.7 |
| (4)  Soft reg. + projection | $59.9 \pm 0.1$ | 62.0 | $52.3 \pm 0.6$ | 55.0 |
| (5)  Soft reg. + proj. + distillation | $59.9 \pm 0.3$ | 62.4 | $52.2 \pm 0.3$ | 54.5 |
| (6)  Soft reg. + proj. + distillation (fine-tuned) | $\mathbf{60.3 \pm 0.1}$ | **62.5** | $\mathbf{52.9 \pm 0.4}$ | **55.2** |
| Entailed questions | | | | |
| (7)  Soft regularizer (symmetric L2 dist.) | $58.6 \pm 0.4$ | 61.3 | $51.5 \pm 0.3$ | 54.3 |
| (8)  Soft regularizer (proposed) | $59.2 \pm 0.2$ | 61.4 | $52.0 \pm 0.5$ | 54.4 |
| (9)  Soft reg. + projection | $59.0 \pm 0.3$ | 60.8 | $52.3 \pm 0.5$ | 54.5 |
| (10) Soft reg. + proj. + distillation | $59.7 \pm 0.2$ | 61.7 | $52.4 \pm 0.3$ | 54.3 |
| (11) Soft reg. + proj. + distillation (fine-tuned) | $\mathbf{59.8 \pm 0.2}$ | **61.7** | $\mathbf{52.5 \pm 0.4}$ | **54.7** |
| Functional programs | | | | |
| (12) Soft regularizer | $59.0 \pm 0.4$ | 61.4 | $51.5 \pm 0.3$ | 53.9 |
| (13) Soft reg. + projection | $58.4 \pm 0.3$ | 61.2 | $50.9 \pm 0.3$ | 53.5 |
| (14) Soft reg. + proj. + distillation | $59.1 \pm 0.3$ | 61.4 | $51.7 \pm 0.6$ | 54.0 |
| (15)  Soft reg. + proj. + distillation (fine-tuned) | $\mathbf{59.4 \pm 0.2}$ | **61.6** | $\mathbf{52.0 \pm 0.5}$ | **54.4** |
| (16) Combination (6) + (11) | $59.9 \pm 0.2$ | 61.9 | $52.8 \pm 0.4$ | 54.9 |
| (17) Combination (6) + (15) | $59.6 \pm 0.2$ | 61.8 | $51.9 \pm 0.4$ | 54.3 |
| (18) Combination (11) + (15) | $59.8 \pm 0.2$ | 62.1 | $52.7 \pm 0.2$ | 54.8 |
| (19) Combination (6) + (11) + (15) | $\mathbf{60.7 \pm 0.1}$ | **62.7** | $\mathbf{53.4 \pm 0.3}$ | **55.7** |

- Annotations of functional programs. The use of these annotations with our method shows a small but positive improvement over the baseline. Again, the soft regularizer is effective on its own, and the additional projection combined with the distillation bring an additional improvement. Remember that we are using here only a limited vocabulary of operations, which is very different from existing methods that rely on complete program annotations to train modular architectures. Our method can use partial annotations and should more easily extend to other datasets and human-produced annotations.

- Combinations. Finally, we combine the best version of each of the three constraints (last rows). The results show that they are complimentary to each other, since the best results are obtained with the full combination. Note however that the relations of equivalence and entailment could, in principle, be deduced from the functional programs. This shows that our handling of program

annotations is still suboptimal. An ideal method should get the full benefits from using program annotations alone.

**Comparison to existing methods**  Finally, we compare our method to existing models using the extended set of metrics proposed in Hudson & Manning (2019) (see Table 3 in the appendix for the full results). Our method performs better than the complex MAC model. More interestingly, we vastly improve on the metric of consistency, which measures the accuracy over sets of related questions about a same image. This is precisely the type of benefit that was expected from our approach.

Finally, in Fig. 3, we plot the accuracy of the model trained on varying amounts of training data, with a baseline and with the proposed method and annotations of equivalence (although a similar trend was observed for the other types of annotations). Very interestingly, the absolute improvement over the baseline is consistent and maintained regardless of the amount of training data used. This strong result suggests that the training signal provided through these additional annotations is truly complementary to the answer annotations, and that this knowledge is otherwise difficult for the model to learn from end-to-end supervision alone. This reinforces the overall motivation for bringing in prior knowledge to the training of deep models.

## 5 CONCLUSIONS

We presented an approach to incorporate prior knowledge in deep learning models in the form of constraints on its internal representations. We then applied the method to the task of VQA to leverage multiple types of relations between training questions. This application is of particular interest because VQA is a prime example of a task where the end-to-end supervised paradigm shows its limits, due to the long chains of reasoning that connect the inputs and outputs. The proposed approach served to shape the space of the internal representations learned by the model. Our experiments with the GQA dataset showed clear benefits in improving the accuracy and robustness of an existing VQA model. Interestingly, these benefits hold regardless of the amount of training data used for end-to-end supervision, suggesting that the type of prior knowledge used cannot otherwise be effectively captured in the model through end-to-end annotations alone. Technically, the proposed method treats the additional knowledge as hard constraints on the model's internal representations. This proved more clearly effective than existing methods to exploit this type of annotations, including soft regularizers and data augmentation.

The proposed method can apply to a variety of tasks and domains that we hope to explore in future work. Concrete applications overlap with those considered for topological embeddings such as learning representations for knowledge bases, and other tree- or graph-structured data (Ganea et al., 2018; Nickel & Kiela, 2017; Vendrov et al., 2016). Enforcing hard constraints on a learned model also allows one to provide guarantees that are otherwise impractical to meet with a purely data-driven approach. In particular, the method could be used to integrate known causal relations in a model, or known outcomes of interventions on specific training examples. This holds the promise of making further steps toward generalizable and trustworthy machine learning models.

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

## A   APPENDIX

## B   IMPLEMENTATION OF THE BASELINE MODEL FOR THE TOY TASK

The hyperparameters and implementation details of the model used for our toy task were set manually for reasonable performance of the baseline model. Most importantly, they were not particularly tuned for optimal performance of the proposed contributions. The size of learned embeddings and all other layers (GRU state, MLP hidden layers) is fixed to 64. We train the model with AdaDelta with mini-batch of size 64. The MLP uses a single layer of gated tanh units (Teney et al., 2018) followed by a linear layer for the regression output. The task loss is the square L2 distance between the model's output and the ground truth value. The ground truth output is always an integer, so we define, as the evaluation metric, the accuracy as the ratio of test instances for which the output rounded to the nearest integer is correct (*i.e.* the predicted output is within less than $0.5$ from the ground truth).

To produce the dataset, we generate every possible instance of the task with a sequence of one to three operations. We reserve 20k of these for validation (*i.e.* model selection and early stopping) and another 20k as our test set. The remaining serves as the training data, of which we use a random fraction depending on the experiment. The code to generate our toy dataset and replicate the experiments will be made available once anonymization issues are cleared.

## C   IMPLEMENTATION OF THE BASELINE MODEL FOR VQA

The VQA model within our method follows the description of Teney et al. (2018). We started with a publicly available implementation of the model. Importantly, all hyperparameters of the BUTD model were first selected for the best baseline performance on the GQA dataset, and were not specifically re-tuned for the contributions of this paper. They were selected by grid search and follow closely Teney et al. (2018).

More specifically, the non-linear operations in the network use gated hyperbolic tangent units. We use the "bottom-up attention" features from Anderson et al. (2018) of size $36 \times 2048$, pre-extracted and provided by Anderson *et al.*[1] The word embeddings are initialized as random vectors (rather than pretrained GloVe vectors) and normalized to a unit L2 norm before being passed to a GRU. We found this to help with the stability of the training and final accuracy, independently from the contributions of our paper. The word embeddings are of dimension 300, and all other layers use embeddings of size 512. The output of the network is passed through a logistic function to produce scores in $[0, 1]$. The final classifier is trained from a random initialization, rather than the visual and text embeddings of Teney et al. (2018). We use Adadelta as the optimization algorithm.

### C.1   TIME COMPLEXITY

The time complexity as a function of dataset size is the same with our method as with standard training. There is a fixed overhead, for each mini-batch, to retrieve the data needed to compute the regularizer (*e.g.* equivalent questions), which is stored alongside the training examples. There is also an additional cost in running the second training phase (distillation). This only involves retraining a few layers for a handful of epochs.

## D   IMPLEMENTATION OF THE PROPOSED APPROACH

Every experiment was repeated four times with different random seeds. We report the mean and standard deviation of answer accuracy of each of the four runs, as well as the result of ensembling the four models by simple averaging of their outputs (predicted scores). Other metrics proposed in Hudson & Manning (2019) are also reported. During training, mini-batches are sampled normally. At most one equivalent and one entailed question (depending on the experiment) are then selected at random, for each question in the mini-batch. Note that many training question do not

---

[1]https://github.com/peteanderson80/bottom-up-attention

have any equivalent/entailed one, while some have several. In the projection by gradient descent in Section 3.2, the number of steps $T$=10 and the step size $\alpha$=0.01. To train networks with distillation, the switch from the first to the second phrase of training occurs when the accuracy on the validation set (test-dev) stops increasing for three epochs.

During the first phase of training, we need to backpropagate the gradient of the loss through the projection function. When using equivalent and entailed questions, the gradient of the projection is trivial. When using functional programs, the projection is a fixed number of steps of gradient descent (see details in the main paper). For the ease of implementation, we approximate its gradient with the identity function.

## E  VQA DATASET

We used the GQA dataset for all of our experiments (Hudson & Manning, 2019). It contains all three types of annotations discussed in this paper. More precisely, for each training question, the dataset provides a set of variable size (possibly empty) of equivalent and entailed questions. It also provides a list of reasoning operations involved in the question. This list uniquely defines the question and vice versa. We use all relations of equivalence. For the relations of entailment, we only keep those involving questions with the same answer. This constitutes most of them, and exceptions are of a form such as *What is the color of the car ? Blue. $\Rightarrow$ Is the car blue ? Yes.* or *Is the kid to the left of the tree ? Yes. $\Rightarrow$ Is the tree to the left of the kid ? Yes.*. To use the annotations of functional programs, we pre-build a vocabulary $\mathcal{O}$ of possible operations. We select operations (unique function/argument combinations) that appear at least 1,000 times in the training set. This corresponds to 354 operations that cover 78% of all annotated operations. All other operations are not used. We use the official balanced training set for training, and the official validation set for hyperparameter tuning and model selection. We used the test-dev and test set for evaluation in tables 2 and 3, respectively.

**Equivalent and entailed questions**   In the annotations provided with the GQA dataset (Hudson & Manning, 2019), a number of pairs of entailed questions are also equivalent questions. We specifically removed those from the entailed questions, such that they form two disjoint sets. Since we propose two techniques for these two types of annotations, it allows evaluating them in isolation. It is also worth mentioning that entailment relations are occasionally dependent on the answer to the premise. For example, the question *Is there a couch or a table that is not brown ?* entails the question *Do you see a white table in picture ?* only if its answer is negative. In our method, constraints are enforced on the embedding of the question alone. However, this does not prevent the whole network to encode answer-dependent relations, since the (constrained) question embeddings further interact with the image and answer representations in the latter layers of the network.

**Vocabulary of operations**   We use annotations of questions as functional programs only with a limited set of operations. The set of all possible operations (combinations of functions and arguments, see (Hudson & Manning, 2019)) is very large, follows a long-tail distribution, and the rare operations will not provide our method with useful information (since it relies on the re-occurrence of operations across multiple questions). To define the vocabulary of operations $\mathcal{O}$ used in our experiments, we selected operations (unique function/argument combinations) that appear at least 1,000 times in the training set. This corresponds to 354 operations that cover 78% of all annotated operations. Annotations of other operations are discarded and not used in our experiments. The threshold of 1,000 was chosen among the values $\{100, 200, 500, 1000, 2000, 4000\}$ for best performance. A larger vocabulary (smaller threshold) did not, but a much smaller vocabulary (higher threshold) clearly decreased the benefit of our method.

Table 3: Additional results: extended metrics on the GQA test set (Hudson & Manning, 2019). Our method consistently improves the consistency metric, which measures agreement across related questions about a same image.

| Method | Accuracy | Open | Binary | Validity | Plausibility | Consistency |
|---|---|---|---|---|---|---|
| Humans | 89.30 | 87.40 | 91.20 | 98.90 | 97.20 | 98.40 |
| CNN+LSTM | 46.55 | 31.80 | 63.26 | 96.02 | 84.25 | 74.57 |
| MAC (Hudson & Manning, 2019) | 54.06 | 38.91 | 71.23 | 96.16 | 84.48 | 81.59 |
| Our BUTD baseline | 53.53 | 38.54 | 70.57 | 96.29 | 84.20 | 83.30 |
| W. equivalent questions (6) | 54.86 | 40.27 | 71.45 | 96.72 | 85.02 | 84.92 |
| W. entailed questions (11) | 54.33 | 39.63 | 71.04 | 96.71 | 84.98 | 85.48 |
| W. functional programs (15) | 54.66 | 40.86 | 70.36 | 96.63 | 84.63 | 85.11 |
| Combination (19) | **55.35** | **40.68** | **72.05** | **96.72** | **84.92** | **87.83** |

