# OpenReview forum: "On Incorporating Semantic Prior Knowlegde in Deep Learning Through Embedding-Space Constraints"
_ICLR.cc/2020/Conference — Reject_

### Official Review · AnonReviewer1 · 2019-10-17
**Official Blind Review #1**

**Rating:** 3

**Review:**

The authors propose a framework to incorporate additional semantic prior knowledge into the traditional training of deep learning models such that the additional knowledge acts as both soft and hard constraints to regularize the embedding space instead of the parameter space. To illustrate the idea, the authors use 3 different annotated knowledge that are already available in a public dataset that contains equivalent statements, entailed statements as well as functional programs and show that the final performance indeed increases.

In general, the paper is well-written and easy to follow. The motivation is clear, i.e., to boost the performance of supervised learning tasks with additional knowledge constraints in a hard way. Compared with the existing models that treat the constraints as soft regularizers, the authors propose to additionally distill the knowledge using teacher-student framework. And this paper contributes in a novel way to incorporate the constraints with both soft and hard training strategies. However, there are several considerations which limits the contribution of this paper:

1. As a teach-student distillation framework, there are several papers using a posterior regularizer with hard constraints, e.g., "Harnessing deep neural networks with logic rules", "Constrained Convolutional Neural Networks for Weakly Supervised Segmentation". More discussions and comparisons with these models should be addressed, and even experimental comparisons if possible, since they also use knowledge distillation to convey the knowledge expressed in the constraints.

2. The proposed model differs with other soft-regularization-based methods in terms of an additional distillation process. The authors state that the combination of task loss with soft regularization lead to over-fitting. To my point of view, the distillation step actually makes similar effect with the case when only optimize the regularizer without the task loss. Hence, I am wondering what's the performance of first using the combined loss and then fix the subsequent layers to only optimize the embedding layers using only the regularization loss. This could demonstrate the difference between the distillation process and the regularization process.

3. Many recent models for VQA have been proposed, e.g, "The Neuro-Symbolic Concept Learner: Interpreting Scenes, Words, and Sentences from Natural Supervision" which also combines extra knowledge as symbolic reasoning. The authors should also compare with such models.

4. It seems the model need to sample a pair of data each time at training to compute the regularizer and also conducting the distillation process. In this case, the time cost should be non-trivial because the distillation process requires optimizing the distance between the current embedding with the hard constraint. Then the question comes as how's the time complexity of the model? What's the convergence speed?

**Experience Assessment:**

I have read many papers in this area.

**Review Assessment: Checking Correctness Of Derivations And Theory:**

I carefully checked the derivations and theory.

**Review Assessment: Checking Correctness Of Experiments:**

I carefully checked the experiments.

**Review Assessment: Thoroughness In Paper Reading:**

I read the paper thoroughly.

---

> ### Author Response · Authors · 2019-11-10
> **Authors' response**
>
> Hi, thanks a lot for your time and valuable comments. They really helped pinpoint sections that required clarifications. We have significatly revised the paper, as summarized below. We believe it to be much improved as a result (updates are highlighted in the PDF).
>
> 1,3> Related works
> - [1] "Harnessing deep neural networks with logic rules"
> This paper used posterior regularization to improve how a learned model complies with hand-designed rules. In comparison, we use *instance-level* auxiliary annotations. They be seen as rules that apply to *some* of the training examples. The scope of the two methods is complementary. The major innovation in our method is to enforce *hard* constraints on the learned embeddings, whereas general rules in [1] are softly balanced with learned predictions. [1] uses teacher/student distillation during training, but their best results are obtained with the teacher network, not with the student. In our case, the distillation step is critical to enforce the hard constraints.
>
> - [2] "Constrained Convolutional Neural Networks for Weakly Supervised Segmentation"
> This one shows how to use image-level tags to learn semantic image segmentation. The tags are turned into linear constraints, which is similar to how we handle program annotations (one of our three use cases). Their contribution is to turn their constrained optimization problem into an objective amenable to SGD that is robust to hard-to-enforce/competing constraints. Our contribution, on the opposite, shows how to enforce constraints strictly. In our particular applications, it proved superior to soft-regularized objectives.
>
> - [3] "The Neuro-Symbolic Concept Learner: Interpreting Scenes, Words, and Sentences from Natural Supervision"
> This paper combines symbolic AI with neural networks for VQA. The vocabulary of visual concepts is learned, but the operation of the model otherwise depends on a pre-specified domain-specific language, and on manually-design modules. They do not require or seek to exploit program annotations for training. Their contributions are essentially orthogonal to ours. Our method could, for example, apply to the representations of questions learned in their semantic parser.
>
>
> 2> Distillation process
> Excellent question: we realized of a gap in the presentation of our motivation for the distillation phase.
> Standard training usually leads to overfitting, which is avoided with early stopping. If one uses a soft regularizer, there is no principled way to design the regularizer to converge right before overfitting occurs.
> With the proposed method, one stops the first training phase once overfitting occurs. Then, the learned regularized embeddings are frozen. We retrain the earlier layers, while the classifier is fixed. The network outputs do not change and there is no further risk of overfitting on the task labels. Two reasons why the second phase succeeds in practice are that (1) we retrain only a handful of layers and (2) we use dense/continuous targets (quite the opposite of training a large-capacity network on sparse labels).
>
>
> 4> Time complexity
> The time complexity (as a function of dataset size) is the same as with standard training.
> - There is a very small fixed overhead, for each mini-batch, to retrieve the data needed to compute the regularizer (e.g. equivalent questions), which is stored alongside the training examples.
> - There is an additional cost in running the second training phase (distillation). This only involves retraining a few layers for a handful of epochs. In our experiments with VQA, we retrain only the question embedding (a word embedding and GRU) for about 5 epochs, whereas the first training phase takes in the order of 20 epochs. The added cost is a very small fraction of the total training time.

---

### Official Review · AnonReviewer2 · 2019-10-21
**Official Blind Review #2**

**Rating:** 6

**Review:**

The paper argues for encoding external knowledge in the (linguistic) embedding layer of a multimodal neural network, as a set of hard constraints. The domain that the method is applied to is VQA, with various relations on the questions translated into hard constraints on the embedding space. A technique which involves distillation is used to satisfy those constraints during learning.

The question of how to encode external knowledge in neural networks is a crucial one, and the limitations of end-to-end learning with supervised data is well-made. Overall I feel that this is a potentially interesting paper, addressing an important question in a novel way, but I found the current version a highly-frustrating read (and I read the paper carefully a number of times); in fact, so frustrating that it is hard for me to recommend acceptance in its current form. More detailed comments below.

Major comments
--
The main problem I have with the paper lies with the first part of section 3, which is a key section describing the main method by which the constraints are satisfied during learning. This is very confusing. The need for the two-step procedure, in particular, and the importance of distillation needs much more explanation, and not relegated to the Appendix (which reviewers are not required to read - see call for papers). I'm not suggesting that the whole of the appendix needs moving to the body of the paper, but I would suggest perhaps 1/2 a page.

A related comment is the use of the distillation technique. This looks crucial, but I don't believe distillation is mentioned at all until the end of the related work section, and even there it comes as a bit of a surprise since there's no mention anywhere of this technique in the introduction.

I would say a little more about the distinction between the embedding space and parameter space, since you say that the external knowledge is encoded in the former and not the latter, and this is important to the overall method. Since embeddings are typically learned (or at least fine-tuned) it's not clear where the boundary is here. Another comment is that embedding space in this paper means the linguistic embedding space. Since this is ICLR and not, eg, ACL, I would make clear what you mean by embedding space.

I don't understand the diagram in Fig. 3 of the architecture, nor the explanation. What's an operation here? Is it *, or *6? I don't get why 3 is embedded by itself in the diagram, and then combined with the remainder using the MLP. Why not just run the RNN over the sequence?

Why are the training instances {3,+1...} and {4,*2,...} equivalent. I stared at this a while, and still have no idea. Also, how are these "known to be equivalent" - what's the procedure?

Minor comments including typos etc.
--
The paper has the potential to be really nicely written and well-presented. Currently it reads like it was thrown together just before the deadline (which only adds to the overall frustration as a reader).

In fig. 1 the second equivalent question example is interesting, since strictly speaking "box" and "rectangular container" are not synonyms (e.g. boxes can be round). Since strict synonymy is hard to find, does that matter? (I realise the dataset already exists and was presented elsewhere, but this might be worth a footnote).

missing (additional) right bracket after Herbert (2016)

Not sure footnote 1 needs to be a footnote. It's already been said, I think, but if it does need repeating it probably deserves to be in the body of the text.

between pairs questions

see Fig.3 -> figure 2?

see Fig.1 -> Tab. 1? (on p.5)

footnote 1 missing a right bracket

usually involve -> involves

+9]) - extraneous bracket

Fig. 4.1 -> Fig. 3? (p.6)

p.7 wastes a lot of space. In order to bring some of the appendix into the main body, I would do away with the very large bulleted list. (I don't mean lose the content - just present it more efficiently)

Remember than

Finally in Fig. 4.2 - some other figure

due of the long chains

**Experience Assessment:**

I have published in this field for several years.

**Review Assessment: Checking Correctness Of Derivations And Theory:**

I assessed the sensibility of the derivations and theory.

**Review Assessment: Checking Correctness Of Experiments:**

I assessed the sensibility of the experiments.

**Review Assessment: Thoroughness In Paper Reading:**

I read the paper thoroughly.

---

> ### Author Response · Authors · 2019-11-10
> **Authors' response**
>
> Hi, thanks for your time and for the excellent quality of the review. We took your comments on-board and did a major revision of the paper, which is now much clearer as a result. Updates are highlighted in the PDF. Thanks a lot for your contribution.
>
> Summary of updates:
> - Introduction: better introduction of technical method, including distillation step.
>
> - Section 3: justification for the 2-phase procedure and for the distillation step; most of technical description moved from the appendix.
>
> - Discussion of constraints in parameter/embedding space. In embedding space, one can constrain how the network represents data. In parameter space, one can guide what the network does with these representations. Both can be useful. The former can be closer semantically to task- or domain-specific knowledge of the data used. Our applications indeed focuses on the *linguistic* embedding space in VQA.
>
> - Fig. 3: thanks for the feedback, this was indeed confusing in many ways. One operation is "*6" or "+3", for example. The input digit and operation sequence are embedded separately to apply constraints on the embeddings of the latter independently. The examples "...+1,*2..." and "...*2,-2,+4..." are marked as equivalent because (x+1)*2 = ((x*2)-2)+4.
> The annotations of equivalent sequences are the assumed "prior knowledge". We made it clearer that the objective was to verify that these annotations bring a useful training signal, complementary to training examples where these sequences are applied on specific x's.
>
> - All typos fixed. LaTeX had shifted all refs to figures and tables.
>
> - Also added a mention of other possible applications in the conclusions (embedding of graph- and tree-structured data).

---

> > ### Comment · AnonReviewer2 · 2019-11-15
> > **Response to new version**
> >
> > Thanks for the changes made to the paper, which makes it much more readable, and hence I will be happy to increase my score accordingly.
> >
> > Just a few typos that have been introduced that may as well get fixed:
> >
> > p.2 typo: hlWe s
> >
> > . Our contribution, on the opposite -> . Our contribution, in contrast
> >
> > that perfectly fit of
> > the desired constraints. - remove "of"

---

### Official Review · AnonReviewer3 · 2019-10-23
**Official Blind Review #3**

**Rating:** 3

**Review:**

This paper proposes the incorporation of “prior knowledge” which enters in the form of the relations between training instances in neural network training. The proposed method is tested on VQA problem, bringing improvements upon the popular soft regularizer.
The authors claim that their method is a general technique but in fact, the constraints are drawn from specific tasks (VQA for example). So, I believe the contribution is rather domain-dependent and not general. Can you explain more how this method can be applied to general problems?

Other than that, I have some concerns:
1. Although the authors claim that they are the first to bring these annotations to VQA, I see their training procedure is closely related to cycle-consistent learning. Recent work in VQA also applied cycle consistency as an online data-augmentation technique (See Shah et al. 2019).
“Shah, M., Chen, X., Rohrbach, M., & Parikh, D. (2019). Cycle-consistency for robust visual question answering.”
2. In Section 2, the authors say “constraints on the parameter space of a model are often non-intuitive”. How are they "non-intuitive" and why the proposed method is more intuitive in terms of theory? Please clarify this.
3. Each question in Hud et al. is associated with a functional program, therefore, questions are compositional. However, arbitrary questions don’t need to strictly follow this constraint. Natural language is not exactly suited to functional programming I think. I have doubts about the claim in Section 4 “Our method can use partial annotations and should more easily extend to other datasets and human-produced annotations”. Also, the definition “A question is defined as a set of operations” does not seem correct. A question can be translated into a program that is composed of a set of operations.
4. Experimental results are not strong enough for such strong claims I believe. Regarding GQA dataset, the authors should compare the proposed method with more works, for example, Hu et al. 2019 and Hudson et al. 2019 achieve much favorable performance upon MAC.
"Hu, R., Rohrbach, A., Darrell, T., & Saenko, K. (2019). Language-Conditioned Graph Networks for Relational Reasoning."
"Hudson, D. A., & Manning, C. D. (2019). Learning by abstraction: The neural state machine."

Minor comments: The paper is not really well written. I even found a wrong reference (Section 3).


**Experience Assessment:**

I have read many papers in this area.

**Review Assessment: Checking Correctness Of Derivations And Theory:**

I assessed the sensibility of the derivations and theory.

**Review Assessment: Checking Correctness Of Experiments:**

I carefully checked the experiments.

**Review Assessment: Thoroughness In Paper Reading:**

I read the paper thoroughly.

---

> ### Author Response · Authors · 2019-11-09
> **Authors' response**
>
> Hi, thanks for the thorough review.
>
> 1> Cycle-consistent learning
> Shah et al. essentially learned a generative model of the question conditioned on the answer, for data augmentation while ensuring that the generated rephrasings lead to the same answer. The only connection with our method is to have multiple examples of a same question, which is really just one of three use cases that we demonstrate. Is there another common aspect that we missed ? I feel that their work and ours are pushing in different directions (they focus on *learning* rephrasings from the data).
>
> 2> Constraints in parameter/embedding space
> In embedding space, one can constrain how the network represents data. In parameter space, one can guide what the network does with these representations. Both can be useful. The former can be closer semantically to a domain expert's knowledge of the data used.
>
> 3> Natural language not compositional
> Excellent point. That is why we did not seek to use the annotations as full program trees, but simply the fact that some questions have some operations in common (e.g. a counting operation, using a 'color' attribute, referring to a 'dog' in the image, etc.). These seem realistic to identify in real questions. Fully agree with the correction "defined"->"translated".
>
> 4> Other methods on GQA
> There are even a few others with higher absolute performance on GQA, but their contributions seem orthogonal to ours. No published method has shown how to benefit from additional annotations as we did. There's nothing in principle that precludes those methods to be combined with our technique. If accepted, we'll certainly include up-to-date references to the state-of-the-art at the time of publication.

---

> ### Author Response · Authors · 2019-11-10
> **Revision uploaded**
>
> We have revised the paper taking into account all points raised in the reviews. We believe that the new version is now much clearer as a result (updates highlighted in the PDF). Thanks again for your contribution !

---

### Decision · Program_Chairs · 2019-12-19

**Decision:**

Reject

**Comment:**

The paper proposes a technique for incorporating prior knowledge as relations between training instances.

The reviewers had a mixed set of concerns, with one common one being an insufficient comparison with / discussion of related work. Some reviewers also found the clarity lacking, but were satisfied with the revision. One reviewer found the claim of the approach being general but only tested and valid for the VQA dataset problematic.

Following the discussion, I recommend rejection at this time, but encourage the authors to take the feedback into account and resubmit to another venue.